# Magnetic and Electrical Characteristics of Nd^3+^-Doped Lead Molybdato-Tungstate Single Crystals

**DOI:** 10.3390/ma16020620

**Published:** 2023-01-09

**Authors:** Bogdan Sawicki, Elżbieta Tomaszewicz, Tadeusz Groń, Monika Oboz, Joachim Kusz, Marek Berkowski

**Affiliations:** 1Institute of Physics, University of Silesia in Katowice, 40-007 Katowice, Poland; 2Faculty of Chemical Technology and Engineering, West Pomeranian University of Technology in Szczecin, 70-310 Szczecin, Poland; 3Institute of Physics, Polish Academy of Sciences, 02-668 Warszawa, Poland

**Keywords:** single crystals, magnetic properties, electrical conductivity, power factor, dielectric spectroscopy

## Abstract

Single crystals of Pb_1−3*x*_▯*_x_*Nd_2*x*_(MoO_4_)_1−3*x*_(WO_4_)_3*x*_ (PNMWO) with scheelite-type structure, where ▯ denotes cationic vacancies, have been successfully grown by the Czochralski method in air and under 1 MPa. This paper presents the results of structural, optical, magnetic and electrical, as well as the broadband dielectric spectroscopy measurements of PNMWO single crystals. Research has shown that replacing diamagnetic Pb^2+^ ions with paramagnetic Nd^3+^ ones, with a content not exceeding 0.01 and possessing a screened 4*f*-shell, revealed a significant effect of orbital diamagnetism and Van Vleck’s paramagnetism, *n*-type electrical conductivity with an activation energy of 0.7 eV in the intrinsic area, a strong increase of the power factor above room temperature for a crystal with *x* = 0.005, constant dielectric value (~30) and loss tangent (~0.01) up to room temperature. The Fermi energy (~0.04 eV) and the Fermi temperature (~500 K) determined from the diffusion component of thermopower showed shallow donor levels.

## 1. Introduction

Divalent metal molybdates and tungstates with the chemical formula of ABO_4_ (A = Ca, Sr, Ba, and Pb; B = Mo or W) and tetragonal scheelite-type structure (space group I4_1_/a), both un-doped as well as activated with RE^3+^ ions, are very interesting materials because of their excellent properties. They are used as efficient phosphors [1], solid state lasers [2,3], optical fibers [4], microwave dielectrics [5,6], catalysts [7,8] and scintillators [9,10,11,12,13]. Recently, other materials with RE^3+^ ions, i.e., their sesquioxides (RE_2_O_3_, RE = Ce, Dy, Gd, Er, Eu, Sm, Yb and Y) are most often used for the fabrication of modern sensors and detectors [14]. In turn, magnetic nanoparticles are now increasingly used in biomedicine [15,16]. Lead molybdate (PbMoO_4_) and lead tungstate (PbWO_4_), as members of the scheelite family, are highly ionic compounds with a small contribution of a covalent bonding. Density functional theory calculations indicated, that both compounds are indirect band gap semiconductors and the values of gap (E_g_) equal to 2.59 eV and 2.96 eV, respectively [17]. However, the experimental values are in the range 3.1–3.2 eV for PbMoO_4_ [18,19] and 3.73–4 eV for PbWO_4_ [20,21].

Our magnetic, electrical, UV-vis diffusion reflection spectroscopy and the broadband dielectric spectroscopy studies of lead tungstate doped with Pr^3+^ ions [20] and lead molybdato-tungstates doped with Pr^3+^ [22] or Gd^3+^ [23,24] ions, have generally shown that they are non-conductive paramagnets or superparamagnets. In particular, microcrystalline samples of Pb_1−3*x*_▯*_x_*Pr_2*x*_WO_4_ with 0.0098 ≤ *x* ≤ 0.20 obtained by high-temperature solid-state reaction method have the indirect band gap for each Pr^3+^-doped sample smaller than E_g_ of pure matrix and it was found to be 3.39 eV (*x* = 0.0098), 3.57 eV (*x* = 0.0839), and 3.42 eV (*x* = 0.20) [20]. In turn, lead molybdato-tungstates doped with Pr^3+^ ions obtained by the same method, i.e., Pb_1−3*x*_▯*_x_*Pr_2*x*_(MoO_4_)_1−3*x*_(WO_4_)_3*x*_ with 0 < *x* ≤ 0.2222, have the maximum value of E_g_ = 3.22 eV for *x* = 0.2, which is even lower than of lead tungstates doped with Pr^3+^ [22]. Interesting results were found for lead molybdato-tungstates doped with Gd^3+^ ions, i.e., Pb_1−3*x*_▯*_x_*Gd_2*x*_(MoO_4_)_1−3*x*_(WO_4_)_3*x*_ materials with *x* = 0.0455, 0.0839, 0.1154, 0.1430, 0.1667 and 0.1774 and *x* = 0.0455, 0.0839, 0.1430 synthesized via solid state reaction route [23] and via combustion one [24], respectively. For the ceramics obtained by the solid state reaction method, it was observed a paramagnetic state with characteristic superparamagnetic-like behavior, the faster and slower dipole relaxation processes up to *x* = 0.1154 and their complete absence above this value [23]. In the materials obtained by combustion route, paramagnetic state with characteristic superparamagnetic-like behavior and the absence of dipole relaxation processes were also observed. This is because the dipole relaxation disappears as the grain size decreases, resulting in a spatial polarization in which the electron or ionic freedom of charge is limited [24].

In the present work, we applied the Czochralski technique to grow scheelite-type Nd^3+^-doped lead molybdato–tungstate single crystals. The growth processes were carried out in air under 1 MPa, which significantly stopped the evaporation of volatile metal oxides. The purpose of our research was to investigate the structural, optical, magnetic and electrical properties of the as-grown single crystals. The novelty of this work is the study of poorly conductive materials that are strongly magnetically diluted. The studies mentioned above allow to determine the influence of magnetic contributions independent of temperature on magnetic parameters. In addition, the Fermi energy and temperature were estimated from the measurements of the diffusion component of the thermoelectric power.

## 2. Materials and Methods

### 2.1. Crystal Growth and Chemical Analysis

Single crystals of Pb_1−3*x*_▯*_x_*Nd_2*x*_(MoO_4_)_1−3*x*_(WO_4_)_3*x*_ solid solution (*x* = 0.001, 0.005 and ▯ denotes vacant sites, labelled later as PNMWO) were successfully grown by the Czochralski method in an inductively heated platinum crucible in air atmosphere under 1 MPa. Starting materials for crystallization processes were the following metal oxides: for PNMWO single crystal (*x* = 0.001)—PbO (109.1223 g; 0.4889 mol; purity 99.95%, Alfa Aesar), MoO_3_ (70.3715 g; 0.4889 mol; purity 99.95%, Alfa Aesar), Nd_2_O_3_ (0.1651 g; 0.4906 mmol; purity 99.99%, Alfa Aesar), and WO_3_ (0.3411 g; 1.4712 mmol; purity 99.95%, Alfa Aesar); for PNMWO single crystal (*x* = 0.005)—PbO (107.8906 g; 0.4834 mol;), MoO_3_ (69.5772 g; 0.4834 mol), Nd_2_O_3_ (0.8256 g; 2.4536 mmol), and WO_3_ (7.3608 g; 7.3608 mmol). The total mass of all oxides used for the pulling of both single crystals was 180.0000 g. The PNMWO single crystals, grown in air atmosphere, are light-brown in color (Figure 1). After accurate orientation by X-ray diffraction, the PNMWO single crystals were cut along the [100] and [001] crystallographic planes, and then plates of dimensions of ~5 × 6 × 1 mm^3^ were cut off. As-prepared samples of PNMWO single crystals were used for the optical, magnetic and dielectric studies. For X-ray diffraction measurements, small pieces of diameters less than 0.1 mm were cut off the as-grown single crystals and selected under a polarization microscope.

The content of neodymium, lead, molybdenum and tungsten in PNMWO single crystals were determined by an Inductively Coupled Plasma Mass Spectrometry (ICP-MS) technique. The contents of metallic elements were found as: for PNMWO single crystal (*x* = 0.001)—Nd 0.08 (1) mas% (cal. 0.08 mas%), W 0.14 (2) mas% (cal. 0.15 mas%), Mo 26.01 (3) mas% (cal. 26.06 mas%), and Pb 56.20 (3) mas% (cal. 56.28 mas%); for PNMWO single crystal (*x* = 0.005)—Nd 0.33 (4) mas% (cal. 0.39 mas%), W 0.68 (3) mas% (cal. 0.75 mas%), Mo 25.66 (5) mas% (cal. 25.76 mas%), and Pb 55.55 (6) mas% (cal. 55.64 mas%). These values closely corresponded to the proposed chemical formula of both crystals.

### 2.2. Methods

Small pieces of the length less than 0.1 mm were cut off from both PNMWO single crystals. The most suitable parts for the single crystal X-ray measurement were chosen under the polarization microscope, and then mounted on a glass capillaries. SuperNova kappa diffractometer, equipped with Mo Kα X-ray tube and Atlas CCD detector (Agilent Technologies), was used for the X-ray diffraction measurements which were performed at 293 (1) K. CrysAlis^Pro^ [25] program was used for collection of the data as well as for determination and refinement of the unit cell parameters from ca. 4000 reflections. Integrations of the collected data were also performed using CrysAlis^Pro^ [25]. SHELXL-97 program [26] was used to refine both structures. The positions of O atoms as well as the anisotropic displacement parameters of all atoms were refined.

Ultraviolet and visible (UV-vis) diffuse reflectance spectroscopy was realized with a JASCO-V670 (Jasco International Co., Tokyo, Japan) spectrophotometer equipped with an integrating sphere. The spectra were recorded in the range from 200 to 1000 nm.

The static (DC) magnetic susceptibility was measured in the temperature range of 5–300 K and recorded both in zero-field-cooled (ZFC) and field-cooled (FC) mode. Magnetization isotherms were measured at 5, 10, 20, 40, 60, and 300 K using a Quantum Design MPMS-XL-7AC SQUID magnetometer (Quantum Design, San Diego, CA, USA) in applied external fields up to 70 kOe. The effective magnetic µ_eff_ moment was determined using the equation [27,28]:(1)$${\;\mu }_{\mathrm{eff}}=\sqrt{\frac{3\mathrm{kC}}{N_{A}\mu_{B}^{2}}}≅2.828\sqrt{C},$$
where k is the Boltzmann constant, N_A_ is the Avogadro number, µ_B_ is the Bohr magneton, and C is the molar Curie constant. The effective number of Bohr magnetons p_eff_ was calculated from the equation:(2)$$p_{\mathrm{eff}}=\sqrt{2{\mathrm{xp}}_{\mathrm{Nd}}^{2}},$$
where $p_{\mathrm{Nd}}=g\sqrt{J\left(J+1\right)}$ [29] for a Nd^3+^ ion (J = 9/2, L = 6, S = 3/2, g = 8/11, basic term ^4^I_9/2_) with 4*f*^3^ electronic configuration.

Electrical conductivity σ(T) of the samples under study was measured by the DC method using a KEITHLEY 6517B Electrometer/High Resistance Meter (Keithley Instruments, LLC, Solon, OH, USA) and within the temperature range of 77–400 K. The thermoelectric power S(T), i.e., the Seebeck coefficient was measured within the temperature range of 100–400 K with the help of a Seebeck Effect Measurement System (MMR Technologies, Inc., San Jose, CA, USA). Dielectric measurements were carried out on PNMWO single crystals which were polished as well as sputtered with (~80 nm) Ag electrodes. The studies were carried out in the frequency range of 5 × 10^2^ – 2 × 10^6^ Hz using a LCR HITESTER (HIOKI 3532–50, Nagano, Japan) and within the temperature range of 80–400 K.

## 3. Results and Discussion

### 3.1. Crystal Structure

The X-ray diffraction measurements revealed that both single crystals belong to tetragonal symmetry and crystallize with scheelite-type structure in I4_1_/a space group, analogously as divalent and scheelite-type molybdates and tungstates, i.e., PbMoO_4_ and PbWO_4_ [17]. The unit cell parameters of PNMWO crystal (*x* = 0.001) are as follows: a = b = 5.4380 (4) and c = 12.1111 (13) Å. The R-value is equal to 0.0149. In the case of PNMWO single crystal (*x* = 0.005) the lattice constants are as follows: a = b = 5.4357 (4) and c = 12.1067 (14) Å. The R-value is equal to 0.0164. The most important crystallographic data are collected in Appendix A.

### 3.2. UV–Vis Diffuse Reflectance Spectra and Optical Band Gap

The optical properties of PNMWO single crystals along both crystallographic directions were investigated at room temperature using UV-vis diffuse reflectance spectroscopy method. The theory which makes possible to use diffuse reflectance spectra for solids was proposed by Kubelka and Munk [30]. According to this method, the reflectance spectra are converted into absorption ones using the following equation [30]:(3)$$F\left(R\right)=\frac{\alpha }{S}=\frac{{\left(1-R\right)}^{2}}{2R}$$ where F(R) is the Kubelka-Munk approach, R is the reflectance, α is the absorption coefficient, and S is the scattering factor which is wavelength independent. Optical band energy gap (E_g_) is related to the absorbance and photon energy the equation proposed by Tauc and Wood [31,32]:(4)$$\alpha h\nu =F\left(R\right)\cdot h\nu =A{\left(h\nu -E_{g}\right)}^{n},$$
where hν is the photon energy, A is an energy independent constant characteristic of a material, and n is a constant that can take different values depending on the nature of electronic transition. The permitted direct, forbidden direct, permitted indirect and forbidden indirect transitions take place when n = 1/2, 3/2, 2 and 3, respectively [31,32]. According to literature information, PbMoO_4_ and PbWO_4_ exhibit the optical spectrum governed by the indirect absorption process, i.e., for n = 2 [33,34]. In the high energy region of the absorption edge, (αhν)^1/2^ varied linearly with photon energy. Thus, in the low energy region, the absorption spectrum deviated from a straight line plot. This straight line behavior in the high energy region was taken as prime evidence of an indirect optical band gap. The plots of (αhν)^1/2^ vs. hν for PNMWO single crystals are depicted in Figure 2. The band gap energy of these crystals are found to be 2.78 eV along the [001] and 2.74 eV along the [100] direction for *x* = 0.001 as well as 2.50 eV along the [001] and 2.44 eV along the [100] direction for *x* = 0.005, corresponding to an indirect permitted transition of an electron between valence and conduction band.

### 3.3. Magnetic Properties

The results of magnetic measurements of PNMWO single crystals are presented in Table 1 and in Figure 3, Figure 4 and Figure 5. Due to the low content of paramagnetic neodymium ions, which did not exceed 0.01, it was necessary to estimate the temperature-independent magnetic contributions, as they affect the magnetic parameters of the single crystals under study. In general, these contributions to susceptibility (χ) coming from the orbital (χ_dia_) and Landau (χ_L_) diamagnetism, Pauli (χ_P_) and Van Vleck (χ_VV_) paramagnetism as well as others, modify the Curie-Weiss law to the form [35]:
(5)$$\chi \left(T\right)=\frac{C}{T-\theta }+\chi_{\mathrm{dia}}+\chi_{L}+\chi_{P}+\chi_{\mathrm{VV}}+\ldots =\frac{C}{T-\theta }+\chi_{0,}$$
where C is the Curie constant, θ is the Curie-Weiss temperature and χ_0_ represents all temperature independent susceptibilities. Multiplying Equation (5) on both sides by the temperature T, we obtain a linear relationship of the product χ⋅T as a function of temperature T in the Curie-Weiss region:(6)$$\chi \cdot T=\frac{C}{1-\frac{\theta }{T}}+\chi_{0}\cdot T=b+\chi_{0}\cdot T,$$
where $b=\frac{C}{1-\frac{\theta }{T}}$ is the intercept that tends to the Curie constant C as the temperature T tends to infinity, i.e., $\underset{T\to \infty }{\lim}\frac{C}{1-\theta /T}=C$, and χ_0_ is the slope. The dependencies of the product χ_ZFC_⋅T(T) from the measurement and the asymptotes determined from Equation (6) in the Curie-Weiss region are shown in Figure 3, and the parameters b and χ_0_ are shown in Table 1

Figure 3 shows that the temperature-independent magnetic contributions are anisotropic, 3a: for the single crystal with *x* = 0.001 and the direction [001], the dominance of Van Vleck’s paramagnetism (χ_0_ > 0) is observed, while for the direction [100]—orbital diamagnetism (χ_0_ < 0), 3b: for the single crystal with *x* = 0.005 and the direction [001], the compensation of Van Vleck’s paramagnetism and orbital diamagnetism (χ_0_ = 0) is observed, while for the direction [100] orbital diamagnetism (χ_0_ < 0) dominates. The χ_0_ values determined from Equation (6) were used to correct the magnetic susceptibility measured in the ZFC and FC modes (Figure 4). This allowed for the correct determination of magnetic parameters such as the Curie constant (C), Curie-Weiss temperature (θ) and the effective magnetic moment (μ_eff_). They are presented in Table 1. It can be seen from Figure 4 that after the χ_0_ correction, both single crystals in both directions are paramagnets with a negative value of the paramagnetic Curie-Weiss temperature, θ (Table 1). This means that short-range magnetic interactions are antiferromagnetic (AFM) in nature. Table 1 shows that the values of the effective magnetic moment (μ_eff_) are significantly higher than the effective number of Bohr magnetons (p_eff_). This may mean that some of the molybdenum ions are in a lower oxidation state than 6^+^, contributing to the total paramagnetic moment. This may explain the existence of short-range AFM interactions, which may result from the competition of interactions between paramagnetic neodymium ions and magnetic molybdenum ones.

Weak paramagnetism is also visible on the magnetization isotherms, M(H), displayed in Figure 5. They showed neither saturation magnetization at 5 K nor magnetic hysteresis for both crystallographic directions resulted in no remanence and coercive field and a transition from paramagnetic to diamagnetic state at 40 K for the single crystal with *x* = 0.001 and at 300 K for the single crystal with *x* = 0.005.

### 3.4. Electrical Studies

The results of the electrical conductivity measurements, σ(10^3^/T), of the PNMWO single crystals clearly showed two areas: extrinsic in the wide temperature range of 77–300 K, in which the weak thermal activation E_a1_ ~0.007 eV is observed as well as intrinsic in the temperature range of 350–400 K with a stronger thermal activation of E_a2_ ~0.7 eV (Table 2 and Figure 6). Despite stronger activation in the intrinsic area, the electrical conductivity value at 400 K is only 1.3 × 10^−3^ S/m. We have low *n*-type electrical conductivity in the intrinsic area (Figure 6 and Figure 7). This behavior correlates well with values of the energy gap in the range of 2.4–2.8 eV, which slightly depend on the content of Nd^3+^ ions in the sample (Table 2). For comparison, the values of E_g_ ~1.7 eV found for both crystallographic directions of CdMoO_4_:Eu^3+^ single crystal [36] are lower than for the single crystals under study, which results in greater electrical conductivity (σ ~9.3 × 10^−3^ S/m) due to the fact that the width of Eu^3+^-multiplet is comparable to the thermal energy kT. Two distinct areas of electrical conductivity with strong activation in the intrinsic region were also observed in ceramics such as: Cu_2_In_3_VO_9_ [37], M_2_FeV_3_O_11_ (M = Mg, Zn, Pb, Co, Ni) [38] and Cd_1−3*x*_Gd_2*x*_▯*_x_*MoO_4_ [39]. The residual *n*-type electrical conductivity in the extrinsic region appears to be related to the anion surplus seen in the chemical formula. Another explanation may be related to the fact that at a state of thermal equilibrium structural defects (n) are always present in the lattice even in the crystal which is ideal in other respects. A necessary condition for free energy minimalization gives: n ≅ Nexp(−E_V_/kT) for n << N, where N is the number of atoms in the crystal, E_V_ is the energy required to transfer the atom from the bulk of the crystal on its surface and k is the Boltzmann constant [40]. On the other hand, *n*-type electrical conductivity in the intrinsic region may be related to the presence of molybdenum ions at a lower oxidation state than 6^+^, similar to magnetic studies, where electrons on the unfilled 4*d* subshell may be a reservoir of current carriers.

The temperature dependence of thermoelectric power, *S*(T), presented in Figure 7, requires special consideration. In general, the thermopower in conventional metals consists of two different parts, i.e., a diffusion component (S_diff_), which according to the Mott formula [41] is proportional to temperature and a phonon resistance component (S_ph_), which is more complex. The S_ph_ contribution results from a transfer of the phonon momentum to the electron gas. It drops both at low temperatures, such as T^3^ below θ_D_/10, when the phonons freeze out (where θ_D_ is the Debye temperature), and at high temperatures, such as T^−1^ above approximately θ_D_/2, when the phonon’s excess momentum is limited by anharmonic phonon-phonon scattering [42]. The Debye temperature θ_D_ [43] has been estimated from the following formula:(7)$$\theta_{D}=\frac{h}{2\pi k}\sqrt[{3}]{\frac{6\pi^{2}N}{V}}v_{D},$$
where h is the Planck constant, N = 24 and V (taken from Appendix A) are the number of ions and the volume of the scheelite unit cell, respectively, and v_D_ = 2692 m/s is the sound speed in PbMoO_4_ matrix [44]. Debye temperature values θ_D_ = 326 K for both crystals and 327 K for the matrix taken from Ref. [44] for comparison. The S(T) minimum observed above 200 K in Figure 7 indicates a transfer of phonon momentum to electron gas.

The diffusion contribution S_diff_ is a direct application of the Boltzmann transport equation [41], as follows:(8)$$S_{\mathrm{diff}}=\frac{\pi^{2}k^{2}}{{\mathrm{eE}}_{F}}T=\mathrm{aT},$$
where e is the elementary charge, E_F_ is the Fermi energy and a is an empirical slope. From Equation (8), the Fermi energy, E_F_, can be written as follows: (9)$$E_{F}=\frac{\pi^{2}k^{2}}{\mathrm{ea}}.$$

Our experimental dependence of S_diff_ on temperature T is marked by solid lines in Figure 7. Equation (9) allows us to evaluate the Fermi energy E_F_ and the Fermi temperature T_F_ (defined as E_F_/k) using the experimental value of the slope of thermopower, a, for each single crystal. The values of E_F_ and T_F_ are summarized in Table 2. Compared to metals, e.g., for pure copper: E_F_ = 7 eV and T_F_ = 8.19 × 10^4^ K [40] and to non-metallic conductors, e.g., for Cu_1−x_Ga_x_Cr_2_Se_4_ single crystals: E_F_ ~0.3 eV and T_F_ ~3 × 10^3^ K [45], the values for materials under study are very small. This means that the Fermi level is near the border of the valence band and the shallow donor level is just below the conduction band. The source of the observed low *n*-type electrical conductivity, which is more thermally activated above room temperature, may be 4*d* electrons derived from molybdenum ions with an oxidation state lower than 6+.

Figure 8 shows an interesting dependence of the power factor S^2^σ on temperature T. The power factor has a very small value of several dozen fW/(cmK^2^). However, its value significantly increases with increasing temperature, i.e., in the intrinsic region above 300 K for a sample richer in neodymium ions, regardless of the crystallographic direction. Similar behavior of S^2^σ(T) with only a few fW/(cmK^2^) was observed in ceramic Gd^3+^ and Co^2+^-co-doped calcium molybdato-tungstates [46] as well as in Nd^3+^ and Mn^2+^-co-doped calcium molybdato-tungstates [47]. The above mentioned studies show that even in ion-bonded materials containing a constant content of 3*d* transition metal ions, thermoelectric efficiency can be improved by doping them with 4*f* rare-earth ions.

### 3.5. Dielectric Properties

Figure 9 presents temperature dependence of the relative dielectric permittivity, ε_r_, for various electric field frequencies of PNMWO single crystals with *x* = 0.001 and 0.005, measured along [001] and [100] crystallographic directions. As can be seen, for each single crystal ε_r_ ~30 remains independent of temperature and frequency up to 300 K and increases rapidly above this temperature as well as at the same time decreases with increasing frequency. In the thermally activated region, the accumulation of electric charge is significantly higher for the sample with *x* = 0.001 (max. ε_r_ ~200) than for the sample with *x* = 0.005 (max. ε_r_ ~100). The loss tangent, tanδ, shows a similar behavior and its value does not exceed 0.01 below room temperature (Figure 10) and a strong energy loss above this temperature, i.e., in a highly thermally activated region. Impedance spectroscopy used to analyze the above results, did not reveal the Cole-Cole semicircles (not presented here). This suggests that in single crystals under study no dipole relaxation processes were observed in the temperature range up to 400 K. Therefore, a charge accumulation visible in the spectra may be caused by the polarization of the space charge in the macroscopic region where the charge freedom of the electron or ion is limited. Similar behavior was found in microcrystalline Gd^3+^-doped lead molybdato-tungstates with the chemical formula of Pb_1−3*x*_▯*_x_*Gd_2*x*_(MoO_4_)_1−3*x*_(WO_4_)_3*x*_ for *x* > 0.1154, synthesized via solid state reaction route [23] and in nanoparticles of the same solid solution obtained via combustion route [24]. This effect was there additionally confirmed by the analysis of the fit of the dielectric loss spectra of Gd^3+^-doped samples by the sum of the conductivity and Havriliak-Negami, Cole-Cole and Cole-Davidson functions [24].

## 4. Conclusions

PNMWO single crystals obtained by the Czochralski method in air and under 1 MPa were characterized by structural, magnetic, UV-vis, electrical conductivity, thermoelectric power, power factor and dielectric spectroscopy measurements. They have shown scheelite-type structure, anisotropic character of the temperature-independent paramagnetic contributions, weak *n*-type conductivity in the extrinsic region (77–300 K) with the activation energy of 0.008 eV and stronger one with the activation energy of 0.7 eV in the intrinsic region (350–400 K), a strong increase of the power factor above room temperature for a crystal with *x* = 0.005, constant dielectric value (~30) and loss tangent (~0.01) up to room temperature. Calculations of the Fermi energy (~0.04 eV) and the Fermi temperature (~500 K) revealed the existence of shallow donor levels. In turn, the impedance spectroscopy analysis showed no dipole relaxation processes and possible charge accumulation through space charge polarization. The final conclusion is that the source of observed weak *n*-type electrical conductivity may be 4*d* electrons derived from molybdenum ions with an oxidation state lower than 6+ and anionic vacancies. Single crystals under study with such properties can be useful in the production of lossless capacitors used in the temperature range up to 300 K.

## Figures and Tables

**Figure 1 materials-16-00620-f001:**
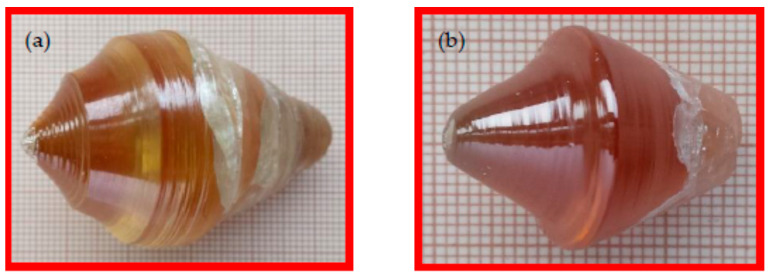
Images of PNMWO single crystals: (**a**) *x* = 0.001 and (**b**) *x* = 0.005.

**Figure 2 materials-16-00620-f002:**
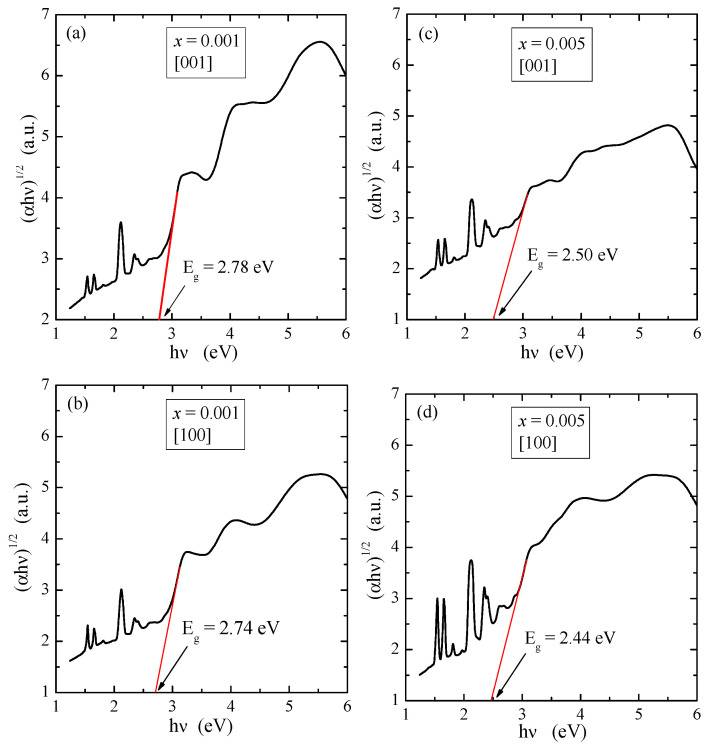
Plots of (αhν)^1/2^ vs. hν of PNMWO single crystals: (**a**) *x* = 0.001 and [001], (**b**) *x* = 0.001 and [100], (**c**) *x* = 0.005 and [001], (**d**) *x* = 0.005 and [100].

**Figure 3 materials-16-00620-f003:**
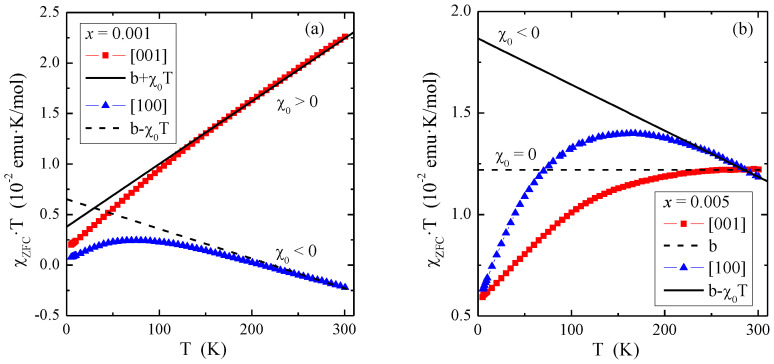
Product χ_ZFC_⋅T vs. temperature T of PNMWO single crystals: (**a**) *x* = 0.001 and (**b**) *x* = 0.005. The solid and dashed lines, χ⋅T(T), indicates Curie-Weiss behavior. χ_0_ is the temperature independent contribution of magnetic susceptibility.

**Figure 4 materials-16-00620-f004:**
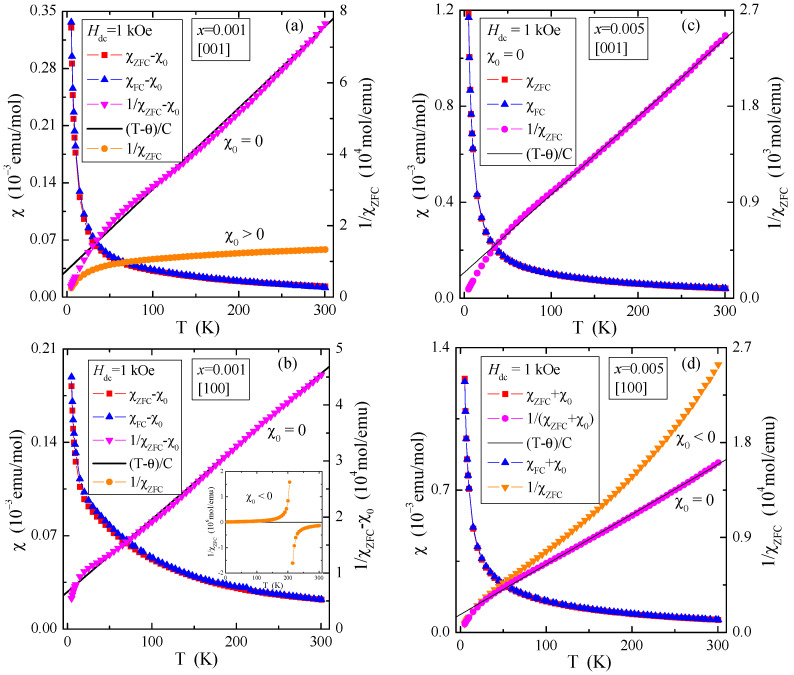
Magnetic susceptibility $\chi_{\mathrm{ZFC}}-\chi_{0}$, $\chi_{\mathrm{FC}}-\chi_{0}$, $1/\left(\chi_{\mathrm{ZFC}}-\chi_{0}\right)$, (T−θ)/C and $1/\chi_{\mathrm{ZFC}}$ vs. temperature T of PNMWO single crystals: (**a**) *x* = 0.001 and [001], (**b**) *x* = 0.001 and [100]. Inset: χ_ZFC_(T) without χ_0_ correction, (**c**) *x* = 0.005 and [001], (**d**) *x* = 0.005 and [100], recorded at *H*_dc_ = 1 kOe.

**Figure 5 materials-16-00620-f005:**
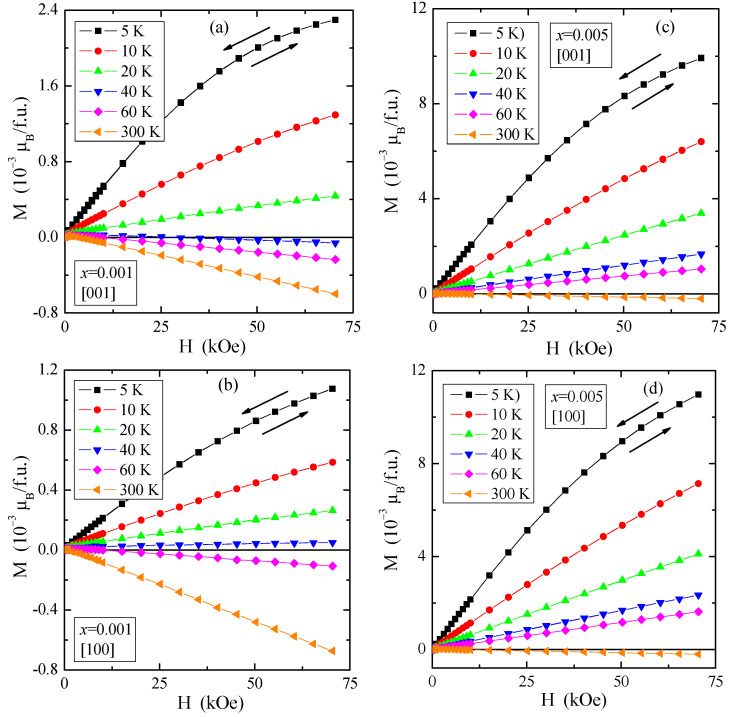
Magnetization M vs. magnetic field H of PNMWO single crystals: (**a**) *x* = 0.001 and [001], (**b**) *x* = 0.001 and [100], (**c**) *x* = 0.005 and [001], (**d**) *x* = 0.005 and [100], recorded at 5, 10, 20, 40, 60 and 300 K. A run of magnetic field is indicated by arrows.

**Figure 6 materials-16-00620-f006:**
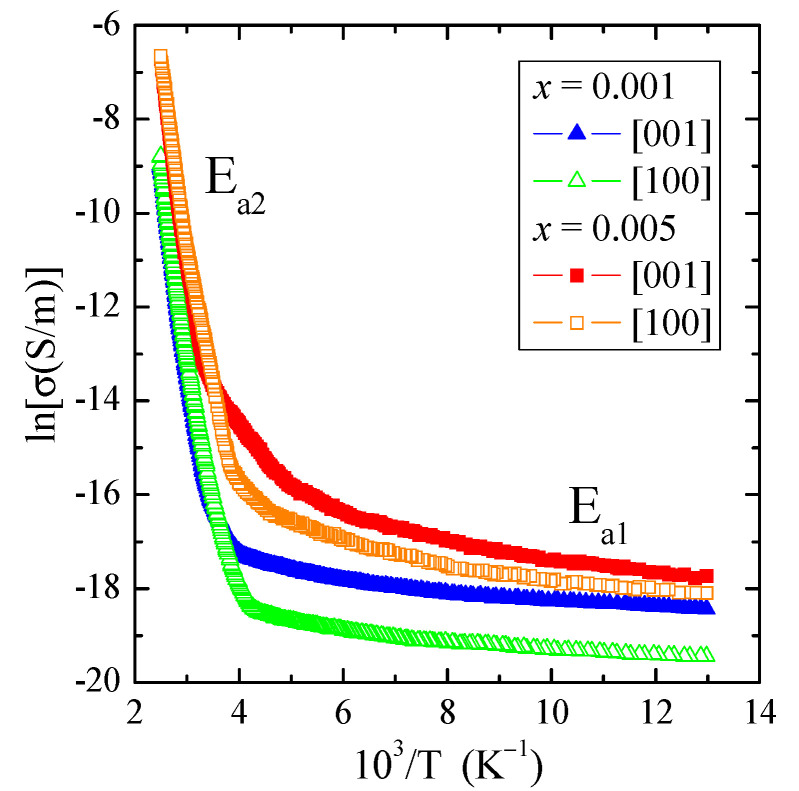
Electrical conductivity (lnσ) vs. reciprocal temperature 10^3^/T of PNMWO single crystals. E_a1_ and E_a2_ are the activation energies in the extrinsic and intrinsic regions, respectively.

**Figure 7 materials-16-00620-f007:**
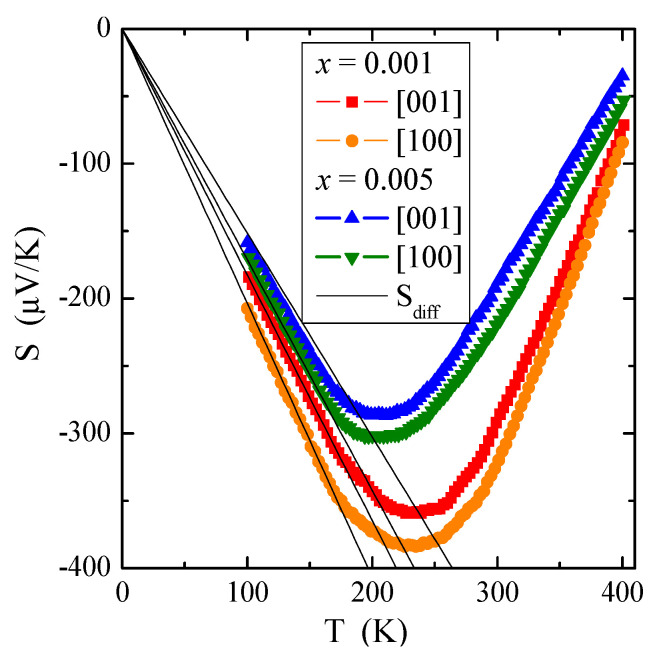
Thermoelectric power S vs. temperature T of PNMWO single crystals. S_diff_ is the diffusion component of thermopower (marked with a solid line).

**Figure 8 materials-16-00620-f008:**
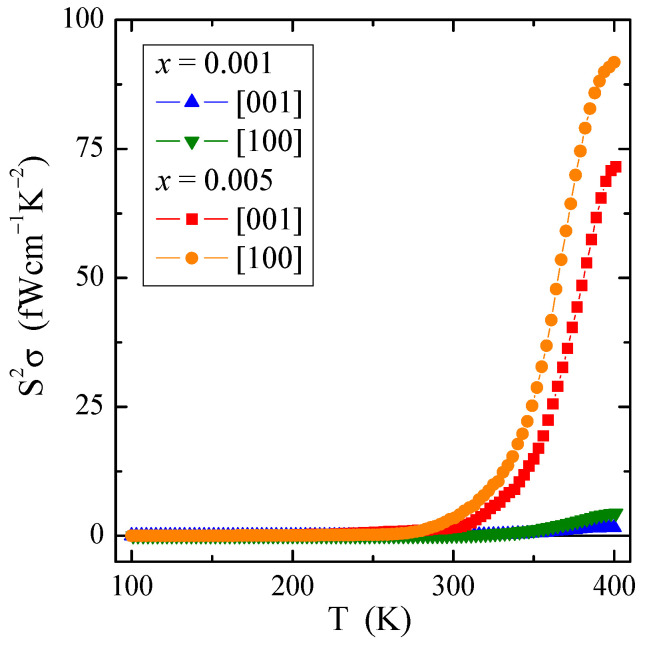
Power factor S^2^σ vs. temperature T of PNMWO single crystals.

**Figure 9 materials-16-00620-f009:**
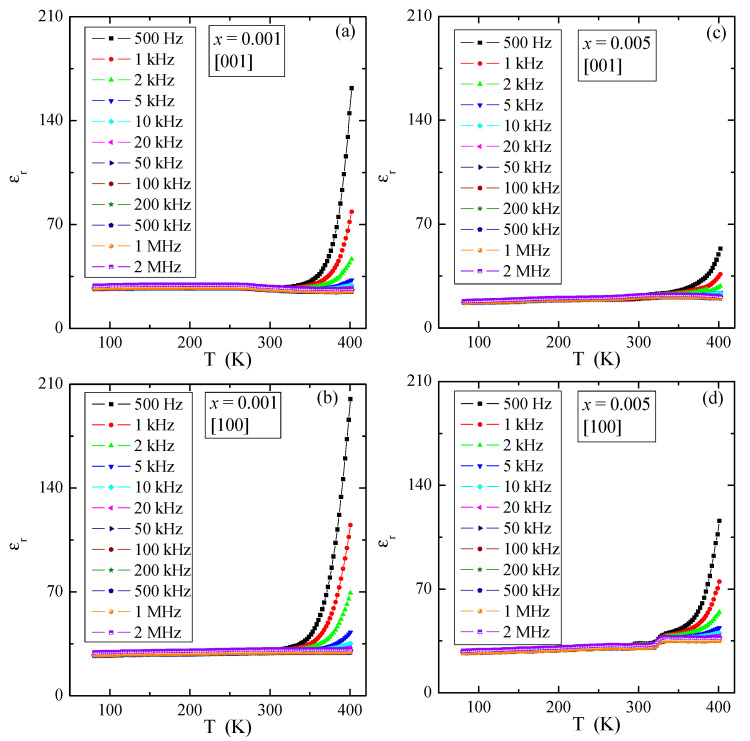
Relative dielectric permittivity ε_r_ vs. temperature T of PNMWO single crystals: (**a**) *x* = 0.001 and [001], (**b**) *x* = 0.001 and [100], (**c**) *x* = 0.005 and [001], (**d**) *x* = 0.005 and [100].

**Figure 10 materials-16-00620-f010:**
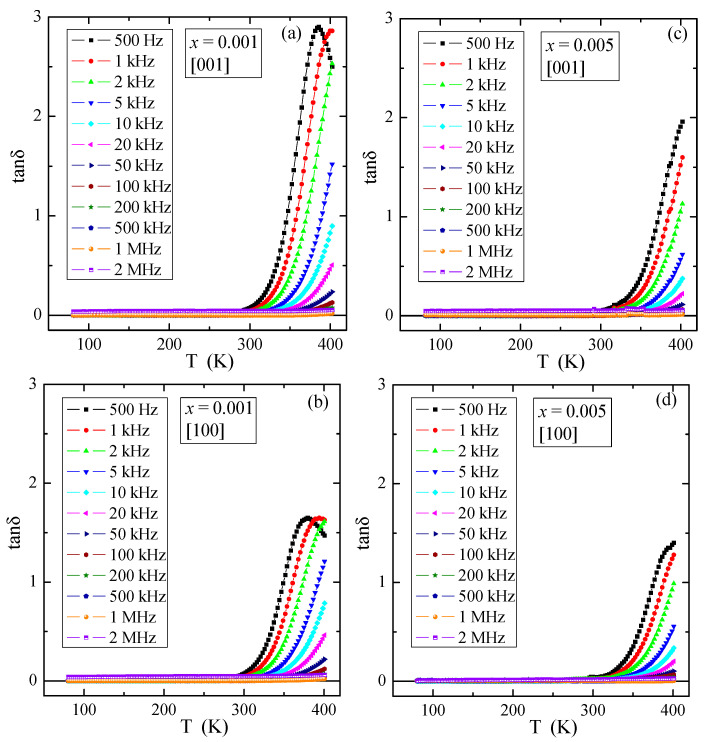
Loss tangent tanδ vs. temperature T of PNMWO single crystals: (**a**) *x* = 0.001 and [001], (**b**) *x* = 0.001 and [100], (**c**) *x* = 0.005 and [001], (**d**) *x* = 0.005 and [100].

**Table 1 materials-16-00620-t001:** Magnetic parameters of PNMWO single crystals.

x	Direction	C (emu⋅K/mol)	θ (K)	µ_eff_ (µ_B_/f.u.)	p_eff_	M_0_ (µ_B_/f.u.)	χ_0_ (emu/mol)	b (emu⋅K/mol)
0.001	[001]	0.0044	−33	0.188	0.114	0.0023	6.2134 × 10^−5^	0.0038
0.001	[100]	0.0077	−51	0.249	0.114	0.0011	−2.9396 × 10^−5^	0.0065
0.005	[001]	0.0137	−33	0.331	0.256	0.0099	0	0.0122
0.005	[100]	0.0211	−36	0.411	0.256	0.0110	−2.2712 × 10^−5^	0.0187

*x* is half of the content of neodymium ions in the crystal, [001] and [100] indicate the plane of the single crystal cut, C is the Curie constant, θ is the Curie-Weiss temperature, µ_eff_ is the effective magnetic moment, M_0_ is the magnetization at 5 K and 70 kOe, p_eff_ is the effective number of Bohr magnetons, χ_0_ and b are the slope and the intercept of the linear χ_ZFC_⋅T(T) function, respectively.

**Table 2 materials-16-00620-t002:** Electrical parameters of PNMWO single crystals.

x	Direction	a (µV/K^2^)	E_F_ (eV)	T_F_ (K)	E_a1_ (eV)	E_a2_ (eV)	E_g_ (eV)
0.001	[001]	−1.824	0.048	557	0.005	0.78	2.78
0.001	[100]	−2.035	0.043	499	0.006	0.72	2.74
0.005	[001]	−1.514	0.040	464	0.011	0.73	2.50
0.005	[100]	−1.712	0.036	418	0.008	0.69	2.44

*x* is half of the content of neodymium ions in the crystal, [100] and [001] indicate the plane of the single crystal cut, a is the slope of the linear S_diff_(T) diffusion function of thermopower, E_F_ is the Fermi energy, T_F_ is the Fermi temperature, E_a1_ and E_a2_ are the activation energies in the intrinsic and extrinsic regions, respectively, and E_g_ is the band energy gap.

## Data Availability

CIF files for PNMWO single crystals have been deposited in the CCDC database with the following Nos. 2221426-2221427.

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
