# Peer review of "Magnetic and Electrical Characteristics of Nd3+-Doped Lead Molybdato-Tungstate Single Crystals"

_materials, 2023, doi:10.3390/ma16020620_

Round 1
Reviewer 1 Report
The study presented in this research is sound, and the results produced are interesting. But a revision is required, and after responding to the following remarks and revising the paper, the manuscript may be considered for publication.
1. Literature review needs to include several recent, relevant publications (high impact) highlighting their key findings. The current version only discussed general aspects while the review of each from several papers is necessary. You may provide a review summary table consisting of a column for the comments or key conclusions.
2. More recent relevant literature or similar work discussion is mandatory in the introduction section, which is missing in the Introduction. Authors are suggested to add one paragraph in the introduction section by discussing the recent progress and citing similar work.
3. The novelty of the work is missing in the introduction. Authors are suggested to include a separate paragraph discussing the novelty and importance of the present work.
4. Authors are suggested to include a literature review on the recent publication on rare-earth doping and magnetic materials and its applications based on the following references in the introduction section: DOIs: 10.1021/acsaelm.1c00703; and 10.1201/9781003197492-10; 10.1016/B978-0-12-823688-8.00002-8.
5. Reduce the similarity. Check the attached similarity report.
6. Also, check the typos throughout the manuscript during revision submission.

Author Response
Detailed response to the Referees’ comments (materials-2094343)
We thank the Reviewers’ for their careful reading of our paper and for their constructive remarks. In order to take into account the latter, the paper has been revised. All changes in the revised manuscript are highlighted in yellow.
Reviewer 1
- General comments:
The study presented in this research is sound, and the results produced are interesting. But a revision is required, and after responding to the following remarks and revising the paper, the manuscript may be considered for publication.
A: The work was corrected according to the Reviewer's comments.
- Detailed comments:
- Literature review needs to include several recent, relevant publications (high impact) highlighting their key findings. The current version only discussed general aspects while the review of each from several papers is necessary. You may provide a review summary table consisting of a column for the comments or key conclusions.
A: The literature review includes 26 out of 42 papers published in the 21st century, to which we include 3 papers suggested by the Reviewer. Our work is not a review but is typical article, it concerns highly magnetically diluted materials.
- More recent relevant literature or similar work discussion is mandatory in the introduction section, which is missing in the Introduction. Authors are suggested to add one paragraph in the introduction section by discussing the recent progress and citing similar work.
A: The following text was added in the Introduction: Recently, other materials with RE3+ ions, i.e. their sesquioxides (RE2O3, RE = Ce, Dy, Gd, Er, Eu, Sm, Yb and Y) are most often used for the fabrication of modern sensors and detectors [14]. In turn, magnetic nanoparticles are now increasingly used in biomedicine [15,16].
- The novelty of the work is missing in the introduction. Authors are suggested to include a separate paragraph discussing the novelty and importance of the present work.
A: The following text was added in the Introduction: In the present work, we applied the Czochralski technique to grow scheelite-type Nd3+-doped lead molybdato–tungstate single crystals. The growth processes were carried out in air under 1 MPa which significantly stopped the evaporation of volatile metal oxides. The purpose of our research was to investigate the structural, optical, magnetic and electrical properties of the as-grown single crystals. The novelty of this work is the study of poorly conductive materials that are strongly magnetically diluted. The studies mentioned above allow to determine the influence of magnetic contributions independent of temperature on magnetic parameters. In addition, the Fermi energy and temperature were estimated from the measurements of the diffusion component of the thermoelectric power.
- Authors are suggested to include a literature review on the recent publication on rare-earth doping and magnetic materials and its applications based on the following references in the introduction section: DOIs: 10.1021/acsaelm.1c00703; and 10.1201/9781003197492-10; 10.1016/B978-0-12-823688-8.00002-8.
A: Suggested publications have been added to the reference list.
- Reduce the similarity. Check the attached similarity report.
A: Similarities have been reduced where possible. Many sentences in the description are standard and "improving" them can lead to a deterioration of their meaning.
- Also, check the typos throughout the manuscript during revision submission.
A: Typos in the manuscript have been checked.

Reviewer 2 Report
Good, solid work, congratulations on high quality single crystals obtained.
The authors should watch for number of self citations, I agree that majority of them seem to be included for a purpose.
Line 86-88: the "Fig 1" tab is inserted in a wrong spot (the pic is of an as-grown crystal, not of a cut-off plate).
Was there any noticeable evaporation of the material at high temperature? Please comment.
The accuracy of weighing is impressive (ex MoO3 (70.3715 g)..... or "total mass ... was 180.0000 g. (?)
The change of lattice constant with the growth direction and across the crystal would be valuable.
Author Response
Detailed response to the Referees’ comments (materials-2094343)
We thank the Reviewers’ for their careful reading of our paper and for their constructive remarks. In order to take into account the latter, the paper has been revised. All changes in the revised manuscript are highlighted in yellow.
Reviewer 2
- General comments:
Good, solid work, congratulations on high quality single crystals obtained. The authors should watch for number of self citations, I agree that majority of them seem to be included for a purpose.
A: We would like to thank the Reviewer for his insightful and high assessment of our work.
- Additional comments:
â—‹ Line 86-88: the "Fig 1" tab is inserted in a wrong spot (the pic is of an as-grown crystal, not of a cut-off plate).
A: This error has been removed.
â—‹ Was there any noticeable evaporation of the material at high temperature? Please comment.
A: Two components of initial reaction mixture, i.e. lead oxide (PbO) as well as molybdenum oxide (MoO3) are relatively volatile compounds. As mentioned in Experimental part, the pulling of both crystals was carried out in air under higher pressure, i.e. under 1 MPa. The higher pressure prevented the evaporation of both oxides. As a result, as-grown single crystals showed the assumed stoichiometric composition.
â—‹ The accuracy of weighing is impressive (ex MoO3 (70.3715 g)..... or "total mass ... was 180.0000 g. (?)
A: The amounts of starting materials were weighed using an analytical balance OHAUS AX324M (maximum measuring range: 320 g; elementary plot: 0.0001 g).
â—‹ The change of lattice constant with the growth direction and across the crystal would be valuable.
A: Lattice constants along the direction of growth and across the crystals have not been studied.

Reviewer 3 Report
The authors present a comprehensive study of the PNMWO single crystals. They were synthesised and well-characterised by a set of methods. Results include determination of structural parameters, optical band gap, magnetic susceptibility and magnetization, electrical conductivity, and Seebeck coefficient. Moreover, Fermi energy is also determined. Conclusions on the physical properties of the system are drawn on the basis of the measurements.
The manuscript is well-written, represents a solid piece of work on the material science, and I recommend it for publication in Materials in the present form.
Author Response
Detailed response to the Referees’ comments (materials-2094343)
We thank the Reviewers’ for their careful reading of our paper and for their constructive remarks. In order to take into account the latter, the paper has been revised. All changes in the revised manuscript are highlighted in yellow.
Reviewer 3
- General comments:
The authors present a comprehensive study of the PNMWO single crystals. They were synthesised and well-characterised by a set of methods. Results include determination of structural parameters, optical band gap, magnetic susceptibility and magnetization, electrical conductivity, and Seebeck coefficient. Moreover, Fermi energy is also determined. Conclusions on the physical properties of the system are drawn on the basis of the measurements. .
A: We would like to thank the Reviewer for his insightful and high assessment of our work.
- Reviewer's recommendation:
The manuscript is well-written, represents a solid piece of work on the material science, and I recommend it for publication in Materials in the present form.
A: Thank you very much for the final positive recommendation from the Reviewer.
